# Elderly People’s Memories about the Itinerary of the HIV/AIDS Diagnosis

**DOI:** 10.3390/geriatrics7050119

**Published:** 2022-10-20

**Authors:** Alessandra de Oliveira, Luana Reis, Arianna Lopes, Elaine Santana, Pollyanna Lima, Thaiza Nobre, Luciana Reis

**Affiliations:** 1Independent School of the Northeast, Collegiate of Nursing, Vitória da Conquista 45.055-030, Brazil; 2Health Sciences Research Unit, Nursing, Nursing School of Coimbra (ESEnfC), 3000-232 Coimbra, Portugal; 3Department of physiotherapy, Federal University of Rio Grande do Norte/FACISA, Natal 59.200-000, Brazil; 4Department of Health I, State University of Southwest Bahia, Jequié 45.208-177, Brazil

**Keywords:** aging, memory, social representations, HIV/AIDS

## Abstract

This study aimed to analyze the remembrances of elderly people about their illness and the path taken in health services until diagnosis of HIV/AIDS. This is a cross-sectional and descriptive analytical study, with a qualitative approach and theoretical and methodological support from the social representations theory and conducted with 38 elderly people. A questionnaire with sociodemographic data, health conditions, and a script for semi-structured interviews was used. Data analysis was performed with the support of the QRS NVivo^®^ software and in light of Bardin’s content analysis. In relation to the itinerary to the diagnosis of HIV/AIDS, the memories are from manifestations of opportunistic diseases and of a long trajectory with health problems and hospitalizations in search of a late diagnosis. The representation of death associated with the diagnosis of HIV/AIDS materializes for some with the loss of the partner in this process of searching for the diagnosis.

## 1. Introduction

Studies have been propagating the increase of sexually transmitted infections (STI) in the elderly, such as human immunodeficiency virus (HIV)/acquired immunodeficiency syndrome (AIDS) [1,2], with an 80% increase in the number of elderly people diagnosed with the disease over the past 14 years. These changes in the epidemiological profile of HIV/AIDS in Brazil reveal the susceptibility of the elderly and demystify the belief in the loss of sexual desire in this group, since most of the infection happens by sexual means, during heterosexual intercourse (heterosexualization) and with reports of unsafe sexual practices, that is, without the use of condoms [3,4].

However, these facts are not the only concerns of researchers in this age group. Added to this is a succession of misfortunes that embody the vulnerability of the elderly to HIV/AIDS infection, such as the false idea of immunity to the disease by this group and the invisibility of their sexuality by professionals, which result in a late diagnosis of HIV/AIDS in the elderly. This favors the onset of opportunistic diseases and other health problems in the elderly, even death [3]. There is also the difficulty in adhering to the use of condoms in sexual relations, especially on the part of men, because they consider themselves non-virile, or even because it is a way of questioning their own fidelity. These behaviors disregard the contagion with HIV, a retrovirus that affects the immune system, a sexually transmitted infection, responsible for developing AIDS [5,6].

The combination of all these factors accentuates the vulnerability of the elderly, not only to infection with HIV/AIDS, but also contributes in a relevant way to a late diagnosis, with complications already installed due to the manifestation of opportunistic diseases and progression of the disease to presentation of AIDS-defining conditions, yet many evolve to death, unaware of the cause and without access to treatment. For this reason, the period that corresponds to the infection with the virus and the revelation of the diagnosis to the elderly person can be long-lasting and full of doubts, fear, numerous visits to health services, and recurrent and prolonged hospital admissions with questioning, with no answer to their illness [3].

It is important to highlight that Brazil has been outstanding in the way of how bravely the country faces the epidemic, through a program that distributes condoms and antiretroviral drugs to all patients free of charge, as well as with the implementation of a public network of laboratories for diagnosis and clinical follow-up of patients and support for research by the Department of Surveillance, Prevention and Control of Sexually Transmitted Infections, HIV/AIDS and Viral Hepatitis, of the Health Surveillance Secretariat, of the Ministry of Health [6].

However, these measures are not enough to stop the spread of the epidemic, but they have demonstrated the capacity to increase survival and improve the quality of life of people living with HIV/AIDS, in addition to enabling the improvement of technical and scientific conditions of health professionals [6].

Although we recognize all the advances and improvements in the access to treatment for those living with the virus, it must be considered that there is a long and difficult way to go for this group, in their pilgrimage in search of diagnosis and care that considers the importance of rebuilding citizenship and social identity for all who live with HIV/AIDS, especially when it comes to older people who, usually, already suffer from social exclusion and the representation of the asexual elderly person.

In the search for giving voice to these social actors who are silenced by the double social stigma of being elderly and living with HIV/AIDS, our goal is to analyze the remembrances of elderly people about their illness and the path taken in health services until diagnosis of HIV/AIDS.

## 2. Material and Methods

This is an analytical-descriptive study, with a qualitative approach, based on the social representations theory. This study was developed in a care center for people with sexually transmitted infections (STIs) and HIV/AIDS in Bahia. The study participants are 38 elderly people, aged 60 years or older, diagnosed with HIV/AIDS, being treated at the HIV/AIDS care center for people with STIs, in the countryside of Bahia, who were contacted/selected through data collection from clinical records and through semi-structured interviews. So, the inclusion criteria adopted was aged 60 years or older and registered in the reference unit. 

The initial number of patients before exclusion was 45 (forty-five) elderly people, of which 7 (seven) were excluded: three did not accept to participate in the research, and four made the appointment but did not show up, even after two new contact attempts.

The data collection tools for this study were used in two stages, on the same day of application: in the first stage, the Mini Mental State Examination (MMSE) was applied, used to exclude elderly people with cognitive deficit. In the second stage, the semi-structured interview script was applied with questions about the disease and diagnosis. The complete MMSE consists of two sections that assess cognitive functions [6]. In the first section, orientation, memory, and attention are assessed, totaling 21 points. In the second section, the ability to name, to obey a verbal and written command, and to copy a complex drawing, in this case a polygon, is evaluated, totaling nine points.

The total score is 30 points, and the cut-off point is 23/24, this being a score suggestive of cognitive deficit [6]. In this sense, the elderly who presented the cut-off point between 23 and 24 in their results were excluded from the study.

First, an initial contact was made with the participants to be interviewed in the waiting room, where they were waiting for treatment; after they approved to participate, the informed consent form (ICF) was handed out and the interviewees’ signatures were collected. Subsequently, the collection instruments and the individual interview were applied. Data collection was performed in a reserved room at the care center, using a mobile device and the KoBoToolbox software (manufacturer Kobo, Inc. Cambridge) [7]. 

From the data collected, we proceeded to the full transcription of the recordings. Then, for the analysis and interpretation of the data collected from the interviews, the content analysis proposed by Laurence Bardin [8] was used, associated with the software QSR NVivo version 12.

For the analysis, it was opted to list the stages of the technique according to Bardin [8], who organizes them in three phases: (1) pre-analysis, (2) exploration of the material, and (3) treatment of the results, inference, and interpretation. Regarding the QRS Nvivo, it is a software that allows you to import and store data. After creating a project in NVivo, you can manage the information through some responsible fields such as sources, nodes and codings, classification, and attributes.

This research project was approved by the Ethics Committee of a Superior Education Institution, under opinion protocol number 3.394.696; after authorization, the data was collected, meeting the fundamental ethical and scientific requirements for research with human beings.

## 3. Results

The elderly people participating in this study have the following characterization: there is a predominance of elderly people with HIV/AIDS who are male (n = 30), in the 60–69 age group (n = 26), separated or divorced (n = 11), with education referring to incomplete elementary school (n = 10) and self-employed as to their profession (n = 16). Regarding the life circumstances in which the diagnosis of HIV/AIDS occurred, n = 12 reported that it was secondary to the partner’s diagnosis (after becoming ill), and another n = 12 through manifestations of opportunistic diseases. As for the most frequent opportunistic diseases, consumptive syndrome (n = 16), tuberculosis (n = 8), and syphilis (n = 5) were cited.

From the analysis of the empirical data, the diagnosis was presented as the preamble to all the narratives, as if, from this diagnosis, a reconstruction of the individual and social identity of the elderly person who receives it had occurred.

Table 1, constructed by the 10 most frequent words in the narratives of the elderly about this process, shows, in a systematic way, the number of times each word appears.

From the analysis made by the characteristics evidenced in the narratives of the elderly people in this study, it is possible to observe that this is an itinerary full of questions, hopelessness, and fear until the diagnosis.

The narratives of (n = 28) older people in this study reveal how this process caused them pain and suffering, and how at this stage they already suffered with some conditions arising from HIV/AIDS, as can be observed in the following speeches:

“I was sick for 3 years and hospitalized for 3 months, it attacked me a lot, I didn’t suspect it. It attacked me like this, I was losing weight, with no appetite to not eat, just losing weight. (Rose 04, 65 years old, married).

I stayed for one year feeling pain, my body was getting weaker, and when I found out, my legs were already paralyzed. I was in the hospital for a long time, I did the exam there, I stayed there 30 days the first time. (Carnation 07, 64 years old, married).

It took two years to find out. I lost a lot of weight, I became thin. I had a lot of diarrhea and syphilis too. (Rose 17, 61 years, in a common-law marriage).”

The data materialized in the testimonies of the participants of this study also reveal in (n = 24) the manifestations of opportunistic diseases, clinical complications, and the level of severity at the time they were diagnosed. For not knowing the real clinical condition, the diagnosis for most participants came from the manifestation of opportunistic diseases, as can be seen in the excerpts described:

I had a lot of diarrhea and pneumonia, I was very thin, so I started the treatment, I was hospitalized for more than 6 months. I had pneumonia and tuberculosis (Carnation 05, 70 years old, widower).

The candidiasis was detected in the endoscopy, before the surgery I had a lot of diarrhea, then I started to suspect, I thought, this business is too strange, it was that watery diarrhea (Carnation 08, 66 years old, separated).

I had tuberculosis. And it was many days that I was hospitalized and without them finding out anything. I stayed more or less three months in the hospital, with bad lungs. I was even in the ICU, I was unconscious (Rose 24, 68 years old, single).

As a result of the worsening of the health condition and the appearance of clinical manifestations related to these opportunistic diseases, hospital admission tends to be a frequent reality, which was mentioned by (n = 10) of the participants, with length of stay varying between 15 days (n = 05), 01 month (n = 03), and 60 days (n = 02), as shown in some of the fragments below: 

I was hospitalized 1 month in a first hospital and 2 months in the second hospital. And... Then I just took medicine, medicine and medicine, I couldn’t even get out of bed, then God gave me strength, then I got better. I went to the dermatologist, I looked for all the doctors, but they didn’t make the diagnosis, right? about the truth of the disease (Rose 04, 65 years old, Married).

I was hospitalized for 21 days. But I wasn’t taking any medication yet (Carnation 06, 69 years old, widower).

I was in the hospital for a long time, I had the exam there in the hospital, I stayed there 30 days the first time (Carnation 07, 64 years old, separated).

The hospitalization, often prolonged due to the criticality of the clinical condition of the elderly person due to the manifestation of the pathology, still without diagnosis, which added to the recurrence of hospitalizations, negatively marks the memories of the itinerary to diagnosis for the elderly in this study. The diagnosis that happens late and incidentally in the hospital setting, in this study (n = 13), with an unfavorable outcome, is often what has collaborated with the statistical data about the higher mortality of this population.

In this context, many fail to access treatment due to the severity, and the diagnosis is followed by therapeutic impossibility, which culminates in the death of the partner, in this study, present in the narrative of (n = 05) participants, as can be recognized in the following narrative:

My second wife died because of it, when the doctors found out there was no way out. She doesn’t even know how she got it, she was from a very noble family, she was hospitalized a couple of times and came out. The third time, the doctors said that was it. And she ended up dying in the hospital (Carnation 13, 71 years old, single).

## 4. Discussion

The data analysis, based on the social representations of the elderly about their therapeutic itinerary, in the health services in search of diagnosis, reveals through the word cloud and the table with the 10 most frequent words in the narratives, where the most evoked words are: fever, test, hospital, and disease which are legitimized in the excerpts of the speeches in which they emphasize the suffering experienced during the process of screening for the disease, the involvement by opportunistic diseases, with hospitalizations and, many times, the diagnosis occurring in the hospital environment and the memory of the losses due to the death of the partner without diagnosis.

Living with HIV/AIDS in old age is living with a double stigma, impacting and capable of modifying one’s experiences, identity, and one’s way of being and behaving in the world [9].

The diagnosis, a passport for inclusion in a new social group to which the elderly does not feel they belong, for the majority makes no sense at all. It occurs after an arduous path, with symptoms and complications that clearly point to a probable destination, but that seems to leave everyone around, even the most skilled professionals, disbelieving of the route, which makes this category in search of diagnosis full of the most varied circumstances. 

Studies point to some etiological factors that drive what they call “epidemiological vulnerability”, which are: the invisibility of the sexuality of the elderly, the end of the reproductive age, and the stigmatization determined by society of this population group [10,11,12,13]. This invisibility is often responsible for a delay in diagnosis, or the lack of it, because they consider it inopportune to investigate sexually transmitted infections in this age group.

The fact is that the social representations built about old age, sexuality, and HIV/AIDS over time remain almost unchanged in contemporary society, not only in speeches, but they are constituent parts of the identity of individuals; therefore, they are present in the actions and behaviors, beliefs, and values of these and disengage them from the approach and information about topics such as: sexuality, sexually transmitted infections, and HIV/AIDS.

According to Moscovici [11], social representations are constructed and shared by public interactions between social actors, in everyday communication practices. These practices and discourses are responsible for the strengthening of taboos and lack of interest in the subject, an unnecessary subject in the perception of professionals and elderly people.

Contrary to what has been practiced in health care by primary health care services, data from the Epidemiological Bulletin-AIDS and STI of 2017 from the Ministry of Health demonstrate that in 2016, when 1294 cases of HIV were registered in the country, there was a 15.0% growth in the rate of people over 60 years old with the virus compared to the previous year [12]. 

The growth of HIV among the elderly population, whether due to the fact that they are aging with HIV/AIDS, or due to the contagion at the age of 60 years or more, requires greater effort and a new look at the diagnosis and welcoming of the elderly in relation to the proposed treatment, in order to ensure adherence and prevent complications associated with the disease, so prevalent in the current scenario of HIV/AIDS diagnosis in the elderly [13].

Because of its biopsychosocial character, sexuality goes beyond the physical/biological realm of the need for reproduction; it is, at the same time, a mix of pleasures ranging from affection and well-being in relationships to cultural and social aspects, passed on from generation to generation [14,15]. Studies corroborate the statement that active sexual life is perpetuated in the elderly phase [3,16]. The elderly often has additional vigor from the support of drug technologies, the use of prosthetics, and increased social opportunities. The refusal of safe sexual practice, due to rejection of condom use, can drive risk behavior to this social group, making them more vulnerable to sexually transmitted infections, including HIV/AIDS [17,18].

Still, the sexuality of the elderly person is an outlawed content in the approach performed by health professionals, by society, in the academic repertoire, and in scientific research. The social representation of an asexual old age is consolidated daily through collective memory, perpetuating non-assistance in such a protocol and homogeneous way that legitimizes the complete exclusion of the approach to sexuality of the aging being, from the interior of social groups to the formulation of public policies and programs of prevention of STIs and HIV/AIDS in primary health care. Thus, they disregard in their scope the necessary visibility capable of guiding the conduct and practices of those who are in charge of care.

For this reason, the elderly remains uninformed or seek information in other ways than through health professionals, as shown by the results of a qualitative study conducted in the south of Brazil, in which all participants denied having addressed, in their consultation with a health professional, information about sexuality, and reported seeking information in the media [19].

In this scenario, despite the growing number of diagnoses of elderly people with HIV/AIDS, late diagnosis has been the reality, followed by manifestations of opportunistic diseases, with complications already installed and, most of the times, with more than one episode of hospitalization [3]. It is noteworthy that, although late, the diagnosis has been made by the secondary or tertiary health care service, materializing the existing gaps in the comprehensive health care of the elderly in the primary service [20].

In this regard, it is emphasized that late diagnosis of HIV in an elderly person favors a higher risk of developing AIDS and complications. However, that is not all: if an elderly person diagnosed late has a CD4 count of less than 200 cells/mL3, they are 14 times more likely to die within the first year of diagnosis [21].

Some researchers have been concerned about the reasons why the elderly are diagnosed with HIV so late, and some findings have been listed, among them: the lack of investigation on sexual activity of the elderly, by health professionals, because they believe that this group is asexual, or by embarrassment and shame to address such content with the elderly during health care [10,19,20]. Thus, consequently, there are serious failures in the prevention of sexually transmitted infections such as HIV/AIDS: the non-request for serological tests, despite symptoms and manifestations of opportunistic diseases, because these symptoms are attributed to the morbidities and declines that affect the elderly and the failure of the health system in relation to information about the disease for this public [10,19,20].

Cruz et al. [21], in their study on late HIV diagnosis in the elderly, warn about the risk factors for the individual and the population of a late diagnosis, such as the delay in starting treatment, the risk of transmission by infected individuals who are unaware of their diagnosis, and the increase in health care costs and morbidity and mortality rates as a result of the disease.

The symptoms presented in the speeches of the elderly are considered signs of the progression of HIV infection, fragility of the immune system, and manifestations of AIDS. The same signs and symptoms were found in other studies and were considered as an indicator condition of HIV/AIDS in patients with late diagnosis, and these are: weight loss, fever, diarrhea, candidiasis, pneumonia, cough, and shingles [22,23].

An epidemiological study of HIV morbidity and mortality conducted in northeastern Brazil between 2013 and 2017 brought evidence that older people living with HIV had higher mortality [24]. This information corroborates data from another study, conducted in Malawi [25], which aimed to assess mortality risk in people living with HIV, stratifying them by age group.

According to the data revealed by this study, the group aged 50 years and older started treatment with a more advanced clinical stage of the disease and a lower CD4 count than the younger ones, and in this age group there was a slower immune response and higher mortality compared to the other groups [25]. This is because the delay in diagnosis, besides allowing the progression of the virus commitment in the organism and manifestations of opportunistic diseases such as pneumonia, tuberculosis, cardiovascular, and neurological impairment, among others, causes a decrease in the effectiveness of antiretroviral therapy (ART) in restoring the immune response [18,26].

All this, added to the chronic diseases, in most cases, already pre-existing, to the natural decline typical of the human aging process, and to the unfavorable socioeconomic issues in which most of the elderly people in Brazil are living, increases the complexity of care, requiring multiple strategies, including medications, and more complex resources [25].

Among developing countries, Brazil was one of the first to ensure universal and free access to ART in the Unified Health System (SUS) in 1996 and has been a reference in the model of care for those living with HIV/AIDS until today, with one of the main objectives to reduce hospital admissions. Therefore, it has outpatient services, day hospital with exclusive laboratories, and a multidisciplinary team to ensure the completeness of care [27].

These measures have proven insufficient in guaranteeing access to services for diagnosis and treatment of HIV/AIDS in specialized services, with professionals qualified for such an initial approach and therapeutic management; thus, the initial contact with the elderly person and/or family members has been performed by professionals in the exercise of their function in hospital care, often in circumstances and acute clinical worsening, without the preparation of the team responsible for transmitting such a clinical finding of such impact to those who receive it, whether the family member or the elderly person.

The Ministry of Health and the World Health Organization guide, by means of protocols, the diagnostic investigation of HIV/AIDS in hospital emergency care in groups with risk behavior and/or patients with symptoms that present themselves among the clinical signs of HIV manifestation. According to the protocol, after diagnostic confirmation, upon discharge from the hospital, the patient should be referred to a basic reference unit or to specialized outpatient clinics for follow-up and investigation for other sexually transmitted infections [6].

It is important to emphasize that the existence of the protocol in non-specialized units is a measure that can help reduce late diagnosis; however, it does not guarantee access to testing or the approach of health professionals for diagnosis, especially when it comes to the elderly, because besides the invisibility of sexuality of this group, already mentioned in this study, the literature points to the embarrassment to talk about the topic, declared by health professionals as another impeditive factor for the professional approach [28,29]. Meanwhile, there is a progressive prolongation and recurrence of hospitalizations, compromising the health of the elderly and the high cost of care provided during hospitalizations, which require multiple technological resources, as in the case of the interviewee who mentioned having spent a week in an intensive care unit.

This qualitative study presents as limitations the impossibility of causal inferences. Thus, the generalization of the results to other municipalities and health services that serve the elderly living with HIV/AIDS would be inconsistent. Therefore, studies focused on the analysis of the care paths taken by the elderly until the diagnosis of infection with the virus are necessary and of humanistic and scientific relevance to guide behaviors and minimize the psychological, physical, and social damage caused by late diagnosis and living with the disease.

## 5. Conclusions

We can infer that the memories of the elderly about their itinerary in the health services until the diagnosis are represented by a vast journey full of questions, suffering, and physical, psychological, emotional, and social decline as a result of hospital admissions secondary to complications arising from opportunistic diseases due to late diagnosis. The representation of death associated with the diagnosis of HIV/AIDS materializes for some with the loss of the partner in this process of searching for the diagnosis.

It is necessary to deepen the discussions about the sexuality of the elderly and it should be a content that is continuously promoted in the academy, during professional training, in scientific research, with the care teams, in society, and within the social groups that compose it.

The grandiosity of the human aging phenomenon brings with it, in the same proportion, not only structural challenges, but also a change of paradigms and a sharp look at all the biopsychosocial issues of the aging being, most of whom are orphans of care in their life trajectory. It is essential that the diagnosis of health problems in the elderly, not only for HIV, be as early as possible, since this group already lives with the decline of the immune system and is more prone to complications and morbidity and mortality, according to the data presented in this study.

## Figures and Tables

**Table 1 geriatrics-07-00119-t001:** Frequency of the 10 most quoted words in the category.

Word	Counting
Fever	23
Test	22
Hospital	21
Disease	19
God	18
Body	13
Weigth	9
Family	8
Revelation	8
Diagnosis	7

Source: research data.

## Data Availability

The data sets of this study are available from the corresponding author upon reasonable request, contactable through: luciana.araujo@uesb.edu.br.

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
