# Peer review of "Elderly People’s Memories about the Itinerary of the HIV/AIDS Diagnosis"

_geriatrics, 2022, doi:10.3390/geriatrics7050119_

Round 1

Reviewer 1 Report

The subject of this manuscript is interesting, but the main problem is the imprecise purpose of the work, the lack of presentation of specific results and the poor consistency of the introduction and discussion with the purpose and results of this study. The manuscript requires major additions and corrections.

Abstract: The purpose should be redrafted and clarified, it should not be the same as the title of the manuscript. Exactly what aspects of the itinerary of the HIV/AIDS diagnosis are analyzed in this study?

The introduction does not explain the rationale and purpose of this study. The research objectives and their importance should be highlighted. Please indicate the gap that this study could fill. Here, too, the purpose should be redrafted.

Line 73. Please provide the initial number of patients before exclusion and the number of excluded.

No characteristics of patients, e.g. age at which HIV / AIDS was diagnosed, disease stage at diagnosis, length of the diagnosis process, sociodemographic factors, etc.

The Results section is a free interpretation of the patients' speeches, the results of the analyzes have not been reported. Please provide how many similar speeches out of 38 registered testify to the indicated result / observation. There is no clear systematics of speeches.

The words in figure 1 should be in English, or please include a legend with a translation.

Table 1 reproduces the results presented in Figure 1.

Discussion: there are no clear references to the results obtained. Certain issues (such as the sexuality of the elderly person) are raised that are poorly related to the results of this work. This section should begin with a brief summary of paper motivations, objectives, and findings. Finally, limitations and future research must be considered.

The conclusions are too general, they should relate to the topic and results of this study.

References: There is no numbering of references, so it is difficult to assess the correctness of citations in the text.

Author Response

Dear reviewers,

Initially, we appreciate your suggestions for corrections as we believe that these suggestions made our article richer. We also confirm that we have complied with all requested corrections.

A hug

Reviewer 2 Report

An analytical-descriptive study was conducted on the manuscript geriatrics-1930171 to assess the importance of late HIV/AIDS diagnosis and memories in manifestations of opportunistic diseases in the elderly. Inclusion criteria and semi-structured interviews should be explained more by the authors. In this way, the reader will know clearly what the subjects are like in this study. Rather than using experimental tests as a means of evaluating and confirming the appearance of disease, why did the authors limit this step to some questions? It would be better if the authors provided some explanations in a table or figure to classify different diagnoses and diseases. Ultimately, a figure or table will make this study easier to understand.

Author Response

(The authors gave the same response as above.)

Round 2

Reviewer 1 Report

I have no further comments.

Reviewer 2 Report

The manuscript is well revised by authors.